# FINE-TUNING CAN HELP DETECT PRETRAINING DATA FROM LARGE LANGUAGE MODELS

**Hengxiang Zhang[1], Songxin Zhang[1], Bingyi Jing[1], Hongxin Wei[1]***

[1]Department of Statistics and Data Science, Southern University of Science and Technology

## ABSTRACT

In the era of large language models (LLMs), detecting pretraining data has been increasingly important due to concerns about fair evaluation and ethical risks. Current methods differentiate members and non-members by designing scoring functions, like Perplexity and Min-k%. However, the diversity and complexity of training data magnifies the difficulty of distinguishing, leading to suboptimal performance in detecting pretraining data. In this paper, we first explore the benefits of unseen data, which can be easily collected after the release of the LLM. We find that the perplexities of LLMs shift differently for members and non-members, after fine-tuning with a small amount of previously unseen data. In light of this, we introduce a novel and effective method termed *Fine-tuned Score Deviation* (**FSD**), which improves the performance of current scoring functions for pretraining data detection. In particular, we propose to measure the deviation distance of current scores after fine-tuning on a small amount of unseen data within the same domain. In effect, using a few unseen data can largely decrease the scores of all non-members, leading to a larger deviation distance than members. Extensive experiments demonstrate the effectiveness of our method, significantly improving the AUC score on common benchmark datasets across various models. Our code is available at https://github.com/ml-stat-Sustech/Fine-tuned-Score-Deviation.

## 1 INTRODUCTION

The impressive performance of large language models (LLMs) arises from large-scale pretraining on massive datasets collected from the internet (Achiam et al., 2023; Touvron et al., 2023b). But, model developers are often reluctant to disclose detailed information about the pretraining datasets, raising significant concerns regarding fair evaluation and ethical risks. Specifically, Recent studies reveal that the pretraining corpus may inadvertently include data from evaluation benchmarks (Sainz et al., 2023; Balloccu et al., 2024), making it difficult to assess the practical capability of LLMs. Besides, LLMs often generate text from copyrighted books (Grynbaum & Mac, 2023) and personal emails (Mozes et al., 2023), which could infringe on the legal rights of the original content creators and violate their privacy. Considering the vast size of the pretraining dataset and the single iteration of pretraining, it has been increasingly important and challenging to detect pretraining data, which determines whether a piece of text is part of the pretraining dataset.

In the literature, current works of detecting pretraining data primarily focus on designing scoring functions to differentiate members (i.e., seen data during pretraining) and non-members (unseen). For example, previous work shows that sequences from the training data tend to have lower perplexity (i.e., higher likelihood) than non-members (Li, 2023). Min-k% leverages the k% of tokens with minimum token probabilities of a text for detection, assuming that trained data tends to contain fewer outlier tokens (Shi et al., 2024). However, non-member data can obtain low perplexities by including frequent or repetitive texts, while members may contain rare tokens that result in high perplexities. This casts significant doubt on utilizing those scoring functions for detecting pretraining data. Consequently, this issue prompts us to present a preliminary attempt to enlarge the difference between members and non-members for pretraining data detection.

---

*Corresponding author (weihx@sustech.edu.cn)

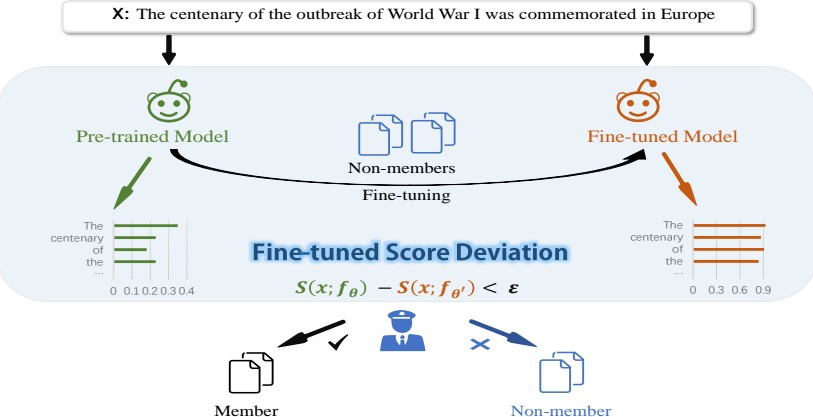

Figure 1: **Overview of Fine-tuned Score Deviation**. Our method first fine-tunes the pre-trained model with a few non-members and then measures the deviation distance of scores from the pre-trained and fine-tuned models as a membership inference metric. If the deviation value is smaller than the threshold value, the text $X$ is likely in the pretraining data.

In this work, we propose Fine-tuned Score Deviation (**FSD**), a novel and effective approach that improves the detection capabilities of current scoring functions in a specific domain (e.g., event data from Wikipedia, arXiv research papers). Our method is motivated by an empirical analysis of the perplexity deviation after model fine-tuning. We find that when fine-tuned with a few previously unseen data from a specific domain, the perplexities of LLMs experience a significantly larger decrease for other *unknown* non-members in the domain compared to the members. This suggests the possibility of using the disparity to distinguish between members and non-members.

Therefore, our key idea behind FSD is to enlarge the score deviation between members and non-members by exposing the LLM to a few non-members. This can be accomplished by measuring the deviation distance of current scores (See Figure 1), owing to the self-supervised fine-tuning on a few non-members. In effect, the fine-tuning largely decreases the scores of non-member data, resulting in more distinguishable seen and unseen data. In practical applications, it is easy to collect a small amount of unseen data for LLMs within a specific domain. For example, we can make use of those contents (e.g., journal articles) published subsequent to the release of the LLM.

To validate the effectiveness of our method, we conduct extensive experiments on various datasets, including WikiMIA, BookMIA (Shi et al., 2024), ArXivTection, BookTection (Duarte et al., 2024) and Pile (Maini et al., 2024). The results demonstrate that our method can significantly improve the performance of existing methods based on scoring functions. For example, our method improves the AUC score of the best baseline method Min-k%, increasing it from 0.62 to 0.91 on WikiMIA under the OPT-6.7B model. Moreover, our method can also improve the TPR@5%FPR score of baseline methods. For example, our method improves the TPR@5%FPR score of the detection method using perplexity, increasing it from 0.10 to 0.81 on ArXivTection under the LLaMA-7B model.

Our main contributions are as follows:

- We analyze the limitations of existing methods based on scoring functions for pretraining data detection. The significant overlap in metric score distribution between seen data and unseen data results in the inferior performance of detection methods.

- We propose Fine-tuned Score Deviation (FSD), a novel and effective method for detecting pretraining data from large language models. The core idea is to enlarge the gap between members and non-members by exposing the LLM to a few unseen data.

- We empirically show that FSD can improve the performance of existing methods based on scoring functions for pretraining data detection, through extensive experiments conducted on various benchmark datasets with diverse LLMs.

## 2 BACKGROUND

In this work, we focus on detecting pretraining data, the problem of detecting whether a piece of text is included in the pretraining data of a specific LLM. First, we formally define the problem setup and its challenges. Then, we introduce two commonly used methods for this task.

**Pretraining data detection**  Pretraining data detection is an instance of membership inference attacks (MIAs) (Shokri et al., 2017), and can help identify data contamination in the pretraining corpus (Shi et al., 2024). Let $f$ be an autoregressive large language model (LLM) with trainable parameters $\boldsymbol{\theta}$ (e.g., LLaMA (Touvron et al., 2023a)) and $\mathcal{D}$ denotes the associated pretraining data, sampled from an underlying distribution $\mathcal{P}$. As model developers rarely provide detailed information about the pretraining dataset, we generally desire to identify if the LLM is trained on the given text for scientific and ethical concerns. Formally, the task objective is to learn a detector $h$ that can infer the membership of an arbitrary data point $\boldsymbol{x}$ in the dataset $\mathcal{D}$: $h(\boldsymbol{x}, f_{\boldsymbol{\theta}}) \rightarrow \{0, 1\}$.

Unlike the black-box assumption in previous works (Shi et al., 2024; Oren et al., 2024), we assume the access to fine-tune LLMs with custom datasets and the output probabilities of LLMs, which is realistic for open-sourced LLMs and many commercial APIs, such as GPT-4o[1]. In addition, the detector can obtain a few data samples $\{\boldsymbol{x_i}\}_{i=0}^{N}$ that belong to the same domain as the given sample $\boldsymbol{x}$ and do not present in the training set. This can be achieved by collecting those contents (e.g., journal articles) published after the release of the LLM.

The task of pretraining data detection can be formulated as a binary classification: determining whether a given text $\boldsymbol{x}$ is a member or non-member of the pretraining dataset $\mathcal{D}$. Pretraining data detection can be performed by a level-set estimation:

$$h(\boldsymbol{x}; f_{\boldsymbol{\theta}}) = \begin{cases} \text{member} & \text{if } \mathcal{S}(\boldsymbol{x}; f_{\boldsymbol{\theta}}) < \epsilon, \\ \text{non-member} & \text{if } \mathcal{S}(\boldsymbol{x}; f_{\boldsymbol{\theta}}) \geq \epsilon, \end{cases} \tag{1}$$

where $\mathcal{S}(\boldsymbol{x}; f_{\boldsymbol{\theta}})$ denotes a scoring function and $\epsilon$ is the threshold determined by a validation dataset. By convention, examples with lower scores $\mathcal{S}(\boldsymbol{x}; f_{\boldsymbol{\theta}})$ are classified as members of pretraining data and vice versa. In the following, we introduce two popular scoring functions for the task.

**Scoring functions**  For large language models, likelihood is typically used to estimate the uncertainty in generating new tokens. In particular, a high likelihood indicates that the model predicts tokens with high confidence. Given a piece of text $\boldsymbol{x} = \{x_1, x_2, ..., x_n\}$, the likelihood of the next token $x_{n+1}$ is $p_{\boldsymbol{\theta}}(x_{n+1}|x_1, ..., x_n)$. In general, a piece of text seen in pre-training tends to have more tokens with a high likelihood, whereas unseen texts have more tokens with a low likelihood.

In light of this, previous studies usually design likelihood-based scoring functions to detect pretraining data (Shi et al., 2024; Carlini et al., 2021; Li, 2023). For example, *Perplexity* is proposed to distinguish members and non-members, based on the observation that members tend to have lower perplexity than non-members (Li, 2023). Formally, The perplexity of $\boldsymbol{x}$ is calculated as:

$$\text{Perplexity}(\boldsymbol{x}; f_{\boldsymbol{\theta}}) = \exp\{-\frac{1}{n}\sum_{i=1}^{n} \log p_{\boldsymbol{\theta}}(x_i \mid x_1, \ldots, x_{i-1})\} \tag{2}$$

where $\boldsymbol{x} = \{x_1, x_2, \ldots, x_n\}$ is a sequence of tokens and $p_{\boldsymbol{\theta}}(x_i \mid x_1, \ldots, x_{i-1})$ is the conditional probability of $x_i$ given the preceding tokens.

Instead of using the likelihood of all tokens, Min-k% (Shi et al., 2024) computes the average probabilities of k% outlier tokens with the smallest predicted probability. The intuition is that a non-member example is more likely to include a few outlier words with low likelihoods than members. Formally, Min-k% is computed by:

$$\text{Min-k\%}(\boldsymbol{x}; f_{\boldsymbol{\theta}}) = \frac{1}{E} \sum_{x_i \in \text{Min-k\%}(\boldsymbol{x})} \log p_{\boldsymbol{\theta}}(x_i \mid x_1, \ldots, x_{i-1}) \tag{3}$$

where $E$ is the size of the Min-k%($\boldsymbol{x}$) set.

---

[1]https://platform.openai.com

However, non-member data can obtain low perplexities by including frequent or repetitive texts, while members may contain rare tokens that result in high perplexities (See Figure 3a and 3b). This issue makes it challenging to distinguish members and non-members using those scoring functions, leading to suboptimal performance in detecting pre-training data. Thus, we present a preliminary attempt to utilize extra non-member data to enlarge the gap between members and non-members.

# 3 METHOD

Recall the realistic assumption that detectors can obtain a few non-members that belong to the same domain as the given sample, we aim to explore how to utilize these extra non-members to improve the detection. Thus, we start by investigating the effects of LLM fine-tuning with unseen examples. Our analysis shows that fine-tuning exerts different impacts on members and non-members.

## 3.1 MOTIVATION

In the analysis, we conduct experiments with WikiMIA (Shi et al., 2024), an evaluation benchmark that uses events added to Wikipedia after specific dates as non-member data. We use $\widetilde{\mathcal{D}}$ to denote the non-member dataset that is accessible for detectors. To construct the dataset $\widetilde{\mathcal{D}}$, we randomly sample a subset with 100 examples from the non-member data of WikiMIA. In addition, we construct the test set with 630 examples each for both members and non-members. Throughout this subsection, we fine-tune LLaMA-7B (Touvron et al., 2023a) with LoRA (Hu et al., 2022) on the non-member dataset $\widetilde{\mathcal{D}}$. To illustrate the effects of fine-tuning, we compare the perplexity distribution of members and non-members from the pre-trained model and the fine-tuned model.

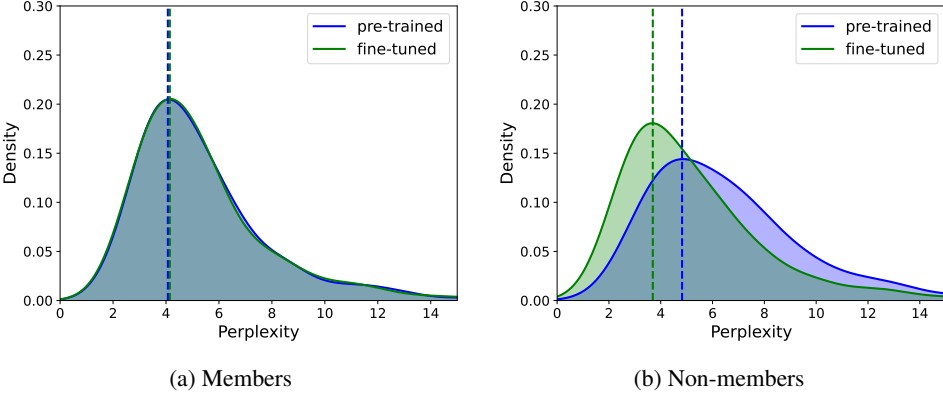

(a) Members        (b) Non-members

Figure 2: The perplexity distribution from the pre-trained model and the fine-tuned model.

**Fine-tuning decreases the perplexity of non-members.** Figures 2a and 2b present the deviation of perplexity distributions for members and non-members, throughout the fine-tuning on the non-member dataset $\widetilde{\mathcal{D}}$. The results show that unseen data in the pretraining tend to obtain a lower perplexity from the fine-tuned model than the pre-trained model. In contrast, we observe that the deviation of the perplexity distribution for members is negligible after fine-tuning. The analysis indicates that fine-tuning with a few unseen data from a specific domain can decrease the likelihood-based scores of the LLM for other unknown non-members in the domain. The contrast in the score deviation resulting from fine-tuning allows for the distinction between members and non-members.

## 3.2 FSD: FINE-TUNED SCORE DEVIATION

Motivated by the previous analysis, we propose Fine-tuned Score Deviation (**FSD**), a general method that can improve the detection performance of current scoring functions in a specific domain. The key idea of our method is to enlarge the gap between seen and unseen data, by exposing the LLM to a few unseen data. With this in mind, we present the details of our approach step by step.

**Construct fine-tuning dataset** Given a piece of text $\boldsymbol{x}$, the first step of our method is to collect a small amount of unseen data for the LLM within the same domain. Owing to the availability of public text data in enormous quantities, we can construct non-member datasets by comparing the LLM release date and data creation timestamp. For instance, we collect some events occurring post-2023 from Wikipedia as the auxiliary non-member dataset for fine-tuning LLaMA (Touvron et al., 2023a), since LLaMA was released in February 2023.

**Fine-tuning with non-members** To expose LLMs to unseen data, we perform fine-tuning on LLMs with the constructed fine-tuning dataset. As our goal is to reduce the perplexity of the unseen data, we employ self-supervised fine-tuning by predicting the next word or token in a given sequence. In particular, we build the loss function by decreasing the negative log likelihood of the actual next token in the sequence. Formally, the fine-tuning loss is:

$$\mathcal{L}_{\text{fine-tuning}}(\boldsymbol{x}) = -\frac{1}{n}\sum_{i=1}^{n}\log f_{\boldsymbol{\theta}}(x_i|x_1, ..., x_{i-1}) \tag{4}$$

**Fine-tuned Score Deviation** Recall that fine-tuning decreases the perplexity of non-members but almost maintains those of members, we propose to exploit the score deviation for detecting pretraining data. Given a new sample $\boldsymbol{x}$, we calculate the score difference between the pre-trained LLM $f_{\boldsymbol{\theta}}$ and the fine-tuned LLM $f_{\boldsymbol{\theta'}}$, where $\boldsymbol{\theta'}$ denotes the parameters of LLM after fine-tuning. Formally, the new score of Fine-tuned Score Deviation (FSD) can be formulated as:

$$\text{FSD}(\boldsymbol{x}; f_{\boldsymbol{\theta}}, f_{\boldsymbol{\theta'}}) = \mathcal{S}(\boldsymbol{x}; f_{\boldsymbol{\theta}}) - \mathcal{S}(\boldsymbol{x}; f_{\boldsymbol{\theta'}}) \tag{5}$$

where $\mathcal{S}(\cdot)$ denotes an existing scoring function, such as Perplexity and Min-k%. With the proposed score, we can estimate the membership of $\boldsymbol{x}$ through the level-set estimation (Eq. (1)). Examples with a large deviation score are considered as non-members and vice versa. In practice, we determine the threshold $\epsilon$ by maximizing detection accuracy on a validation set, following the previous work (Shi et al., 2024). Our method is compatible with various scoring functions and consistently enhances their performance in detecting pretraining data, as presented in Table 1.

By way of the FSD score, we can obtain a clear distinction between members and non-members, establishing excellent performance in detecting pretraining data. To provide a straightforward view, we show in Figure 3 the score distribution between members and non-members using various scoring functions on WikiMIA (Shi et al., 2024). The results of ArXivTection (Duarte et al., 2024) are also presented in Appendix D.1. Our experiments validate that, compared to the perplexity and Min-k% scores, our FSD score significantly increases the gap between non-members and members, and as a result, enables more effective pretraining data detection.

## 4 EXPERIMENTS

In this section, we evaluate the effectiveness of our method for pretraining data detection across several benchmark datasets with multiple existing open-sourced models. We also apply FSD to copyrighted book detection in real-world scenarios and find it a consistently effective solution.

### 4.1 EXPERIMENTAL SETUP

**Models** We conduct extensive experiments on diverse open-sourced LLMs. For the main results, we use LLaMA-7B (Touvron et al., 2023a) as the LLM throughout our experiments. We also provide experiments on other models including Pythia-6.9B (Biderman et al., 2023), GPT-J-6B (Wang & Komatsuzaki, 2021), OPT-6.7B (Zhang et al., 2022), LLaMA-13B models (Touvron et al., 2023a), LLaMA-30B (Touvron et al., 2023a), and NeoX-20B (Black et al., 2022). Existing works (Shi et al., 2024; Ye et al., 2024) generally use these models as LLMs for performing the studies of pretraining data detection. The models are provided by Hugging Face [2].

**Datasets** To verify the effectiveness of detection methods, we employ four common benchmark datasets for evaluations, including WikiMIA (Shi et al., 2024), ArXivTection (Duarte et al., 2024),

---

[2] https://huggingface.co/models

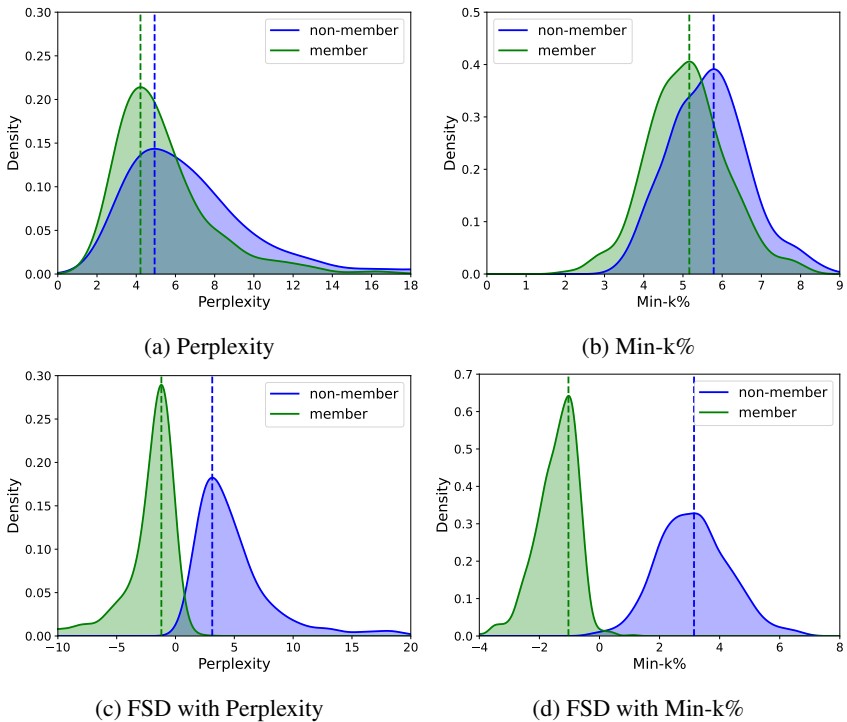

Figure 3: Distribution of scores from pre-trained model vs. FSD. We contrast the distribution of scores from the pre-trained model using perplexity and our FSD with perplexity(a & c). Similarly, we contrast the Min-k% scores distribution from the pre-trained model and our FSD (b & d). Using FSD leads to enlarging the gap between members and non-members.

BookTection (Duarte et al., 2024) BookMIA (Shi et al., 2024) and Pile (Maini et al., 2024). Previous works have demonstrated that model developers commonly use text content among those datasets for pre-training (Shi et al., 2024; Duarte et al., 2024; Ye et al., 2024). The datasets are provided by Hugging Face[3], and detailed information of datasets is presented in Appendix B.

**Baseline methods**  We use four detection methods based on scoring functions as baselines to evaluate the performance of different methods across various datasets and models. Those methods employ specific metrics based on the likelihood, followed by a comparison with a preset threshold to identify the membership of the given text. Specifically, baseline methods include the example perplexity (**Perplexity**) (Li, 2023), the ratio of example perplexity and zlib compression entropy (**Zlib**) (Carlini et al., 2021), the ratio of the perplexity on the example before and after lowercasing (**Lowercase**) (Carlini et al., 2021) and detecting pretraining example through outlier words with low probability (**Min-k%**) (Shi et al., 2024).

**Evaluation metrics**  We evaluate the performance of different detection methods for detecting pre-training data from large language models, by measuring the following metrics: (1) AUC, the area under the receiver operating characteristic curve; (2) the true positive rate (TPR) when the false positive rate (FPR) of the examples is 5% (TPR@5%FPR).

**Implementation details**  Our approach involves constructing the non-member dataset to fine-tune the base model. For constructing the non-member dataset, we randomly sample 30% of the data from the entire dataset and select all non-members from this subset as the constructed fine-tuning dataset. The remaining 70% of the dataset is used for testing. We employ LoRA (Hu et al., 2022) to fine-tune the base model with 3 epochs and a batch size of 8. We set the initial learning rate as

---

[3] https://huggingface.co/datasets

Table 1: AUC score for pretraining data detection with baselines and our method from various models on WikiMIA and ArXivTection. *Base* and *+Ours* respectively denote the baseline methods and our method. **Bold** shows the superior result.

| Dataset | Method | GPT-J-6B | | OPT-6.7B | | Pythia-6.9B | | LLaMA-7B | | NeoX-20B | |
|---|---|---|---|---|---|---|---|---|---|---|---|
| | | *Base* | *+Ours* | *Base* | *+Ours* | *Base* | *+Ours* | *Base* | *+Ours* | *Base* | *+Ours* |
| WikiMIA | Perplexity | 0.64 | **0.95** | 0.60 | **0.90** | 0.64 | **0.90** | 0.64 | **0.92** | 0.69 | **0.93** |
| | Lowercase | 0.59 | **0.77** | 0.59 | **0.71** | 0.58 | **0.74** | 0.58 | **0.69** | 0.66 | **0.76** |
| | Zlib | 0.61 | **0.94** | 0.59 | **0.89** | 0.61 | **0.88** | 0.62 | **0.90** | 0.64 | **0.93** |
| | Min-k% | 0.68 | **0.92** | 0.62 | **0.91** | 0.67 | **0.86** | 0.65 | **0.85** | 0.73 | **0.90** |
| ArXivTection | Perplexity | 0.79 | **0.96** | 0.68 | **0.89** | 0.77 | **0.95** | 0.68 | **0.92** | 0.79 | **0.95** |
| | Lowercase | 0.59 | **0.81** | 0.58 | **0.70** | 0.60 | **0.77** | 0.50 | **0.69** | 0.62 | **0.75** |
| | Zlib | 0.64 | **0.96** | 0.55 | **0.89** | 0.63 | **0.95** | 0.57 | **0.91** | 0.65 | **0.95** |
| | Min-k% | 0.85 | **0.92** | 0.74 | **0.84** | 0.84 | **0.91** | 0.76 | **0.86** | 0.85 | **0.91** |

Table 2: The average AUC score of baselines and our method from the Pythia-6.9B over 20 subsets of the Pile dataset. *Base* and *+Ours* respectively denote the baseline methods and our method.

| Method | Perplexity | | Lowercase | | Zlib | | Min-k% | |
|---|---|---|---|---|---|---|---|---|
| | *Base* | *+Ours* | *Base* | *+Ours* | *Base* | *+Ours* | *Base* | *+Ours* |
| Pile | 0.503 | **0.625** | 0.519 | **0.566** | 0.507 | **0.624** | 0.515 | **0.600** |

0.001 and drop it by cosine scheduling strategy. We conduct all experiments on NVIDIA L40 GPU and implement all methods with default parameters using PyTorch (Paszke et al., 2019).

## 4.2 MAIN RESULTS

**Can FSD improve the performance of current scoring functions?** We compare the performance of detection methods on WikiMIA and ArXivTection datasets across various LLMs. The details of the dataset split are shown in Appendix C.1. The results in Table 1 show that the FSD significantly improves the performance of all baselines on both datasets across diverse models. For example, our method improves the AUC score compared to the best baseline method Min-k%, increasing it from 0.62 to 0.91 on the WikiMIA dataset under the OPT-6.7B model. Similarly, it improves the AUC score from 0.76 to 0.86 on the ArXivTection dataset under the LLaMA-7B model. Moreover, our method improves the TPR@5%FPR score of all baselines, which can be found in Appendix D.2. We also present the results on different subsets of the Pile dataset under the Pythia-6.9B model. Table 2 shows the average AUC scores of baselines and our method over the 20 subsets of Pile, demonstrating that our method achieves superior performance on the Pile dataset. In addition, the detailed results of all subsets of the Pile dataset are provided in Appendix D.3.

**How does the fine-tuning data size affect the performance of FSD?** To investigate the effect of varying the fine-tuning data size on the pretraining data detection, we compare the performance of our method using different-sized fine-tuned datasets for fine-tuning LLaMA-7B. To construct fine-tuning datasets of varying sizes, we randomly sample varying amounts of non-members (0, 30, 50, 100, 150, 200, 250, 300) from the WikiMIA dataset as fine-tuning datasets. In addition, we construct a balanced test set of 930 examples for evaluation.

Figure 4 presents the performance of FSD with various sizes of auxiliary datasets. The results show our method achieves better performance as the size of the fine-tuning dataset increases. Notably, our method is highly data-efficient, achieving dramatic improvements with only a small amount of non-members for fine-tuning. For example, FSD improves the AUC score of the perplexity-based method from 0.63 to 0.91, by leveraging only 100 non-member data for fine-tuning – a significant direct improvement of 44%. In summary, a few non-members are sufficient for FSD to improve the detection, demonstrating its practicality. In addition, we also evaluate our method on the BookC2 subset of the Pile dataset under the Pythia-6.9B model. The results show a similar trend, which can be found in Appendix D.3.

Table 3: AUC score for pretraining data detection with baselines and our method from the different-sized LLaMA model on WikiMIA. *Base* and **+Ours** respectively denote the baseline methods and our method. **Bold** shows the superior result.

| Method | LLaMA-7B | | LLaMA-13B | | LLaMA-30B | |
|---|---|---|---|---|---|---|
| | *Base* | *+Ours* | *Base* | *+Ours* | *Base* | *+Ours* |
| Perplexity | 0.64 | **0.92** | 0.66 | **0.92** | 0.68 | **0.91** |
| Lowercase | 0.58 | **0.69** | 0.60 | **0.70** | 0.60 | **0.75** |
| Zlib | 0.62 | **0.90** | 0.63 | **0.90** | 0.65 | **0.91** |
| MIN-K% | 0.65 | **0.85** | 0.67 | **0.86** | 0.70 | **0.82** |

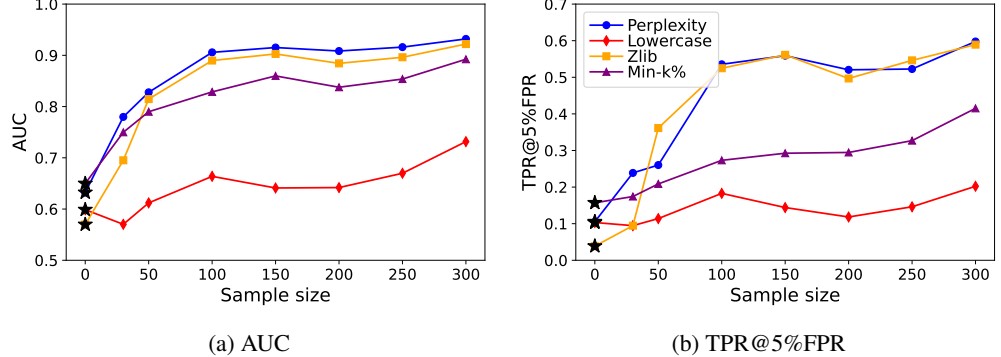

(a) AUC            (b) TPR@5%FPR

Figure 4: AUC and TPR@5%FPR of scoring functions with FSD, using auxiliary datasets with varying sizes. Notably, ★ represents the baseline without FSD.

**Is FSD effective with different-sized models?** We also verify the performance of baselines and our methods from different-sized LLaMA models (7B, 13B, 30B) on WikiMIA. In Table 3, our results demonstrate that our method is effective with different-sized models, and achieves remarkable performance from different-sized models. Notably, the AUC score of **Lowercase** slightly rises as the parameters of the LLaMA model increase. Moreover, additional results of the TPR@5%FPR score show a similar trend, which can be found in Appendix D.2.

**Can our method detect copyrighted books in pretraining data?** Recent works (Shi et al., 2024; Duarte et al., 2024) study the problem of copyrighted book detection in training data. Following previous works, we verify the effectiveness of detection methods for detecting excerpts from copyrighted books on BookMIA (Shi et al., 2024) and BookTection (Duarte et al., 2024). Specifically, we randomly sample 500 members and non-members from datasets, constructing a balanced validation set of 1,000 examples. The detailed information of datasets split is presented in Appendix C.2.

In Table 4, we compare the accuracy of our method and baselines for detecting suspicious books in pretraining data from the LLaMA-7B model. A salient observation is that our method significantly improves the accuracy of baseline methods for copyrighted book detection. For example, compared with baselines, our method reaches an accuracy of 98.6% on BookMIA using detection method **Zlib**, which marks a significant 71.8% improvement. We also present the AUC score with our method and baselines, which consistently improves the detection capabilities of baseline methods. Our extensive experiments demonstrate the effectiveness of our method for copyrighted book detection.

## 5 DISCUSSION

**Can members be used for fine-tuning?** The key step of our method is to fine-tune the pre-trained model using a few non-members. One may also ask: *can a similar effect be achieved by utilizing members as the fine-tuning dataset?* In this ablation, we separately sample members and non-members from WikiMIA to construct fine-tuning datasets(Mem, Non). In addition, we randomly sample data from WikiMIA as another fine-tuning dataset (All). The details of implementation are

Table 4: Accuracy and AUC score for copyrighted book detection with baselines and our method from LLaMA-7B on BookTection and BookMIA. *Base* and *+Ours* respectively denote baslines and our method. **Bold** shows the superior result.

| Metric | Accuracy | | | | AUC | | | |
|--------|----------|--|--|--|-----|--|--|--|
| Method | BookTection | | BookMIA | | BookTection | | BookMIA | |
| | *Base* | *+Ours* | *Base* | *+Ours* | *Base* | *+Ours* | *Base* | *+Ours* |
| Perplexity | 66.9 | **85.4** | 59.0 | **96.5** | 0.710 | **0.910** | 0.564 | **0.995** |
| Lowercase | 64.5 | **73.0** | 67.0 | **69.2** | 0.664 | **0.770** | 0.708 | **0.779** |
| Zlib | 65.3 | **86.4** | 57.4 | **98.6** | 0.568 | **0.920** | 0.474 | **0.999** |
| MIN-K% | 68.1 | **82.1** | 59.5 | **93.9** | 0.716 | **0.880** | 0.587 | **0.979** |

Table 5: AUC of scoring functions with FSD, using members (Mem), non-members (Non), and mix of them (All) on LLaMA-7B. Base denotes the scoring function without FSD. **Bold** shows the superior result.

| Method | Base | All | Mem | Non |
|--------|------|-----|-----|-----|
| Perplexity | 0.64 | 0.68 | 0.78 | **0.92** |
| Lowercase | 0.58 | 0.54 | 0.67 | **0.69** |
| Zlib | 0.62 | 0.65 | 0.79 | **0.90** |
| MIN-K% | 0.65 | 0.61 | 0.81 | **0.85** |

Table 6: AUC of scoring functions with FSD using the original WikiMIA, data removing timestamps (Deletion), and data replacing the year of timestamps with 2023 (Replacement). The results are shown as Base/+Ours.

| Method | WikiMIA | Deletion | Replacement |
|--------|---------|----------|-------------|
| Perplexity | 0.64/ **0.92** | 0.62/ **0.76** | 0.54/ **0.71** |
| Lowercase | 0.58/ **0.69** | 0.58/ **0.62** | 0.52/ **0.63** |
| Zlib | 0.62/ **0.90** | 0.58/ **0.72** | 0.55/ **0.68** |
| MIN-K% | 0.65/**0.85** | 0.61/ **0.69** | 0.54/ **0.67** |

presented in Appendix C.3. To investigate the impact of different fine-tuning datasets on pretraining data detection, we fine-tune the LLaMA-7B model with each fine-tuning dataset.

Our results in Table 5 show that our method can improve the performance of baseline methods using members as the fine-tuning dataset. However, our method performs inferior when using members for fine-tuning compared with non-members. Moreover, it is not realistic to construct a member dataset without accessing pretraining data in real-world scenarios. In addition, this is feasible for constructing non-members as a fine-tuning dataset based on the model release date and data creation timestamp. Overall, our method achieves superior performance when using non-members for fine-tuning, while ensuring applicability in real-world settings. We also investigate the performance of our method when fine-tuning using data from an unrelated domain. The results in Appendix D.3 show that our method can improve the performance when fine-tuning data from a mix of domains.

**Is our method affected by distribution difference?** Existing works generally construct benchmark datasets based on the LLM release date and data creation timestamp (Ye et al., 2024; Shi et al., 2024). For example, the WikiMIA dataset considers events occurring post-2023 as non-members. Recent works indicate evaluation results are suspect on benchmark datasets because they possibly sample members and non-members from different distributions (Duan et al., 2024; Das et al., 2024; Maini et al., 2024). We find the temporal shift between members and non-members in the WikiMIA dataset, which is shown in Appendix C.4. The issue shows that we may distinguish members and non-members with timestamps in the dataset. To eliminate the impact of temporal differences between members and non-members on evaluation, we implement two strategies to mitigate the temporal shift in the dataset: (1) removing timestamps in the dataset (Deletion), and (2) replacing the year of timestamps with 2023 in the dataset (Replacement). We conduct experiments with baselines and our method on the original WikiMIA dataset, Deletion and Replacement, respectively.

Our results in Table 6 show that our method also improves the performance of baseline methods when mitigating the temporal shift between members and non-members. In this setting, the performance of baselines and our method is reduced, as deleting or replacing a word will change the probability of the subsequent word, thereby perturbing the likelihood-based metric. Though baseline methods yield results comparable to random guessing on the Replacement dataset, our method improves the AUC scores of the perplexity-based method, increasing it from 0.54 to 0.71 on Replace-

Table 7: AUC score of FSD with different fine-tuning methods. Base denotes baseline methods without model fine-tuning. **Bold** shows the superior result.

| Metric | AUC | | | | TPR@5%FPR | | | |
|---|---|---|---|---|---|---|---|---|
| Method | Base | AdaLoRA | IA3 | LoRA | Base | AdaLoRA | IA3 | LoRA |
| Perplexity | 0.64 | 0.82 | 0.91 | **0.92** | 0.09 | 0.39 | **0.52** | 0.41 |
| Lowercase | 0.58 | 0.62 | **0.72** | 0.69 | 0.10 | 0.13 | 0.17 | **0.18** |
| Zlib | 0.62 | 0.76 | 0.84 | **0.90** | 0.09 | 0.24 | 0.32 | **0.47** |
| Min-k% | 0.65 | 0.80 | **0.90** | 0.85 | 0.15 | 0.22 | **0.39** | 0.25 |

ment. Overall, our method is effective even if there is no distribution difference between members and non-members. The TPR@5%FPR score of the experiment is presented in Appendix D.2.

**Is FSD effective with different fine-tuning methods?**    To expose LLMs to unseen data, we employ LoRA to fine-tune the pre-trained model. The results demonstrate that our method achieves impressive performance for pretraining data detection when fine-tuning with LoRA. However, can a similar effect be achieved using different fine-tuning methods? To this end, we investigate the impact of fine-tuning methods on performance by applying AdaLoRA (Zhang et al., 2023), IA3 (Liu et al., 2022), and LoRA to fine-tune LLaMA-7B with WikiMIA, respectively.

In Table 7, we report the AUC and TPR@5%FPR scores for pretraining data detection for our method and baselines. The results show that our method improves the performance of baselines when using different fine-tuning methods. Although our FSD achieves inferior performance with AdaLoRA compared with IA3 and LoRA, it still improves the performance of baseline methods. Our method can be implemented with different fine-tuning methods and does not require a specific fine-tuning technique. In addition, we also conduct experiments to explore the impact of different fine-tuning parameters on the performance of our method. The results show that our method is insensitive to LoRA rank and the number of fine-tuning epochs, which are presented in Appendix D.3.

## 6 CONCLUSION

In this paper, we introduce Fine-tuned Score Deviation (**FSD**), a novel detection method that can universally improve the performance of existing detection methods. To the best of our knowledge, our method is the first to utilize some collected non-members in the task of pretraining data detection. Our core idea behind FSD is to enlarge the gap between seen examples and unseen examples by exposing the LLM to a few unseen examples. In effect, unseen data have a larger score than seen examples when using FSD, which makes it more distinguishable between seen and unseen data. Extensive experiments demonstrate the effectiveness of our method for detecting pretraining data on common benchmark datasets across various models. In summary, the FSD is an effective approach for accurately detecting pretraining data of LLMs.

**Limitations**    Our method requires to collect a few examples that belong to the same domain but are not involved in the training. Generally, we can utilize the data content published after the release of the LLM. Therefore, our method is applicable for detecting benchmarks or copyrighted resources in a specific domain (e.g., math tests, magazines). The diversity of the test set may make it challenging to construct an effective auxiliary dataset of unseen data. In addition, our method requires fine-tuning on a few non-member data, so the effectiveness of the proposed score might be affected by the data quality of non-members.

**Ethical Statement**    Our work focuses on pretraining data detection from large language models. The proposed methodology aims to address issues involving data contamination or copyright infringement. In addition, our method can be used to identify potential privacy leakage risks and ensure the safety of LLMs, aligning with established ethical standards for content moderation. Regarding data access, the evaluated datasets we employed in our work come from prior research and do not involve personal privacy information.

ACKNOWLEDGEMENT

This research is supported by the Shenzhen Fundamental Research Program (Grant No. JCYJ20230807091809020) and the SUSTech-NUS Joint Research Program. Bingyi Jing is supported in part by the National Natural Science Foundation of China under grant 12371290. We gratefully acknowledge the support of the Center for Computational Science and Engineering at the Southern University of Science and Technology.

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

## A    RELATED WORK

Pretraining data detection, which is an increasingly important topic for large language models, relates to a large amount of literature on membership inference attacks and data contamination. We discuss some of the relevant works to ours in two directions below.

**Membership Inference Attacks**   Our work mainly studies how to detect a given example in the pretraining data, which is consistent with the objective of membership inference attacks (MIAs) (Shokri et al., 2017; Truex et al., 2019). This task aims to determine whether a given data point is a member of training data. Metric-based attack methods, such as loss (Yeom et al., 2018), entropy (Salem et al., 2019), confidence (Liu et al., 2019) and gradient (Liu et al., 2023), infer membership of data by comparing the calculated metric value with a preset threshold. Previous works have generalized metric-based methods to large language models (Duan et al., 2024; Xie et al., 2024; Zhang et al., 2024; Mattern et al., 2023), by calculating the based-likelihood metric (e.g., perplexity) for membership inference. Recent works apply MIAs to pretraining data detection by designing likelihood-based scoring functions to measure the membership of data (Shi et al., 2024; Ye et al., 2024). In this work, we analyze the limitations of existing scoring functions for pretraining data detection, and design an effective method to improve their performance. In particular, this work is the first to explore the importance of collecting unseen data in pretraining data detection.

**Data Contamination**   Data contamination has been studied in the literature (Xu et al., 2024a; Magar & Schwartz, 2022; Balloccu et al., 2024), where training data may inadvertently include evaluation benchmark data, resulting in unauthentic evaluation results. Thus, it is important to assess the leakage of benchmark data into pretraining data (Zhou et al., 2023). On the one hand, model developers can remove evaluation benchmark data from training data by retrieval-based methods with access to pertaining data (Ravaut et al., 2024; Chowdhery et al., 2023). Specifically, those methods employ n-gram tokenization and string-matching for detecting data contamination (Brown et al., 2020; Touvron et al., 2023b; Team et al., 2023; Radford et al., 2019). On the other hand, researchers utilize prompting techniques (Golchin & Surdeanu, 2024), performance analysis (Ye et al., 2024; Debenedetti et al., 2024), model likelihood (Oren et al., 2024; Shi et al., 2024; Xu et al., 2024b) to detect potential contamination without access to the training data. Our work focuses on pretraining data detection, an area that is similar to data contamination. Different from data contamination detection, our FSD can also be applied to the detection of copyrighted resources in real-world scenarios.

## B    DETAILS OF DATASETS

Previous works construct benchmark datasets to evaluate the performance of detection methods for pretraining data detection. Following the prior literature, we conduct experiments on 5 benchmark datasets: WikiMIA (Shi et al., 2024) selects old Wikipedia event data as member data by leveraging the Wikipedia data timestamp and the model release date, since Wikipedia is a commonly pretraining data source. BookMIA (Shi et al., 2024), which contains excerpts from copyrighted books in the Books3 subset of the Pile dataset (Gao et al., 2020), can be used for detecting potential copyright infringement in training data. ArXivTection (Duarte et al., 2024) is a curated collection of research articles sourced from ArXiv. BookTection (Duarte et al., 2024), which comprises passages from 165 books, is constructed based on BookMIA. We also conducted experiments on the Pile dataset (Maini et al., 2024), which is large-scale text dataset for training language models, including text data from various sources such as books, GitHub, and website content.

## C    EXPERIMENTAL DETAIL

### C.1    DATASET SPLIT

We report the performance of detection methods on WikiMIA and ArXivTection datasets across various large language models. In our experiments, we first randomly sample 30% of the dataset, and then select all non-members from this subset to construct the fine-tuning dataset. The remaining

Table 8: The train set and test set used in the experiment

| Dataset | Type | Member | Non-member | Total |
|---------|------|--------|------------|-------|
| WikiMIA | Train Set | \ | 231 | 231 |
| | Test Set | 599 | 558 | 1,157 |
| ArXivTection | Train Set | \ | 238 | 238 |
| | Test Set | 536 | 549 | 1,085 |

Table 9: The train set, test set and validation set used in the experiment

| Dataset | Type | Member | Non-member | Total |
|---------|------|--------|------------|-------|
| BookMIA | Train Set | \ | 1,413 | 1,413 |
| | Test Set | 2,887 | 3,022 | 5,909 |
| | Validation set | 500 | 500 | 1,000 |
| BookTection | Train Set | \ | 1,796 | 1,796 |
| | Test Set | 6,833 | 3,657 | 10,490 |
| | Validation set | 500 | 500 | 1,000 |

70% of the dataset is used for testing. The detailed information of the constructed dataset is shown in Table 8.

## C.2 COPYRIGHTED BOOK DETECTION

To conduct experiments of copyrighted book detection on BookMIA and BookTection, we randomly sample 30% of the dataset and select all non-members from this subset as the fine-tuning dataset. Subsequently, we randomly sample 500 members and non-members from the remaining 70% of the datasets, constructing a balanced validation set of 1,000 examples for evaluation. The detailed information dataset split is shown in Table 9.

## C.3 FINE-TUNING WITH MEMBERS

To investigate the impact of model fine-tuning with different fine-tuning datasets on pretraining data detection, we construct three types of fine-tuning datasets. In this ablation, we sample members (**Mem**) and non-members (**Non**) from WikiMIA as fine-tuning datasets, respectively. In addition, we randomly sample data from WikiMIA to construct a fine-tuning dataset (**All**). The details of fine-tuning datasets are shown in Table 10

Table 10: The train set and test set used in the experiment

| Datasets | Type | Member | Non-member | Total |
|----------|------|--------|------------|-------|
| Mem | Train Set | 262 | \ | 262 |
| | Test Set | 599 | 558 | 1,157 |
| Non | Train Set | \ | 231 | 231 |
| | Test Set | 599 | 558 | 1,157 |
| All | Train Set | 262 | 231 | 493 |
| | Test Set | 599 | 558 | 1,157 |

## C.4 TEMPORAL SHIFT

In Table 11, we provide some illustrations of the temporal shift between members and non-members in the WikiMIA dataset.

Table 11: Illustrations of temporal shift between the member and non-member distributions.

| Members | Non-Members |
|---|---|
| The 2014 On 30 June or 2 July 2014, the Armed Forces of the Democratic Republic of the Congo and United Nations forces launched an offensive against rebel groups in the Masisi and Walikale. | The 95th Academy Awards was a ceremony held by the Academy of Motion Picture Arts and Sciences (AMPAS) on March 12, 2023, at the Dolby Theatre in Los Angeles. |
| In 2014, a series of groundbreaking diplomatic meetings was held between Wang Yu-chi, in his official capacity as the Minister of the Mainland Affairs Council (MAC) of the Republic of China (ROC). | The 36th Annual Nickelodeon Kids' Choice Awards ceremony was held on March 4, 2023, at the Microsoft Theater in Los Angeles, California with Nate Burleson and Charli D'Amelio. |
| Concluding observations on the second periodic report of the Holy See was a 2014 report issued by the Office of the United Nations High Commissioner for Human Rights. | The 2023 Summer Metro Manila Film Festival is an ongoing iteration of the annual Summer Metro Manila Film Festival held in Metro Manila and throughout the Philippines. |
| The 2014 European Aquatics Championships took place from 13 to 24 August 2014 in Berlin, Germany. | On February 11, 2023, an octagonal unidentified flying object was detected over northern Montana. |
| The centenary of the outbreak of World War I was commemorated in Europe in late July and early August 2014. | The 2023 Tokyo Marathon was the 16th edition of the annual marathon race in Tokyo, held on Sunday, 5 March 2023. |

Table 12: TPR@5%FPR score for pretraining data detection with baselines and our method from various models on WikiMIA and ArXivTection. *Base* and *+Ours* respectively denote the baseline methods and our method. Bold shows the superior result.

| Dataset | Method | GPT-J-6B | | OPT-6.7B | | Pythia-6.9B | | LLaMA-7B | | NeoX-20B | |
|---|---|---|---|---|---|---|---|---|---|---|---|
| | | *Base* | *+Ours* | *Base* | *+Ours* | *Base* | *+Ours* | *Base* | *+Ours* | *Base* | *+Ours* |
| WikiMIA | Perplexity | 0.12 | **0.78** | 0.12 | **0.63** | 0.13 | **0.66** | 0.09 | **0.41** | 0.20 | **0.58** |
| | Lowercase | 0.12 | **0.24** | 0.07 | **0.18** | 0.11 | **0.25** | 0.10 | **0.18** | 0.16 | **0.18** |
| | Zlib | 0.09 | **0.78** | 0.09 | **0.55** | 0.10 | **0.50** | 0.09 | **0.47** | 0.10 | **0.57** |
| | MIN-K% | 0.17 | **0.40** | 0.14 | **0.50** | 0.17 | **0.35** | 0.15 | **0.25** | 0.25 | **0.36** |
| ArXivTection | Perplexity | 0.26 | **0.79** | 0.12 | **0.63** | 0.25 | **0.66** | 0.10 | **0.81** | 0.27 | **0.77** |
| | Lowercase | 0.13 | **0.23** | 0.15 | **0.22** | 0.15 | **0.25** | 0.09 | **0.16** | 0.13 | **0.20** |
| | Zlib | 0.15 | **0.80** | 0.07 | **0.60** | 0.14 | **0.50** | 0.08 | **0.66** | 0.16 | **0.77** |
| | MIN-K% | 0.42 | **0.57** | 0.24 | **0.45** | 0.41 | **0.35** | 0.24 | **0.45** | 0.40 | **0.58** |

## D  DETAILED EXPERIMENTAL RESULTS

### D.1  FINE-TUNED SCORE DEVIATION

We show in Figure 5 the score distribution between members and non-members using two scoring functions on ArXivTection. The results also demonstrate that our FSD score significantly increases the gap between non-members and members compared to the perplexity and Min-k% scores, thus enabling more effective pretraining data detection.

### D.2  DETAILED RESULTS OF EXPERIMENT

We report the TPR@5%FPR score for pertaining data detection in Table 12, 13, 14.

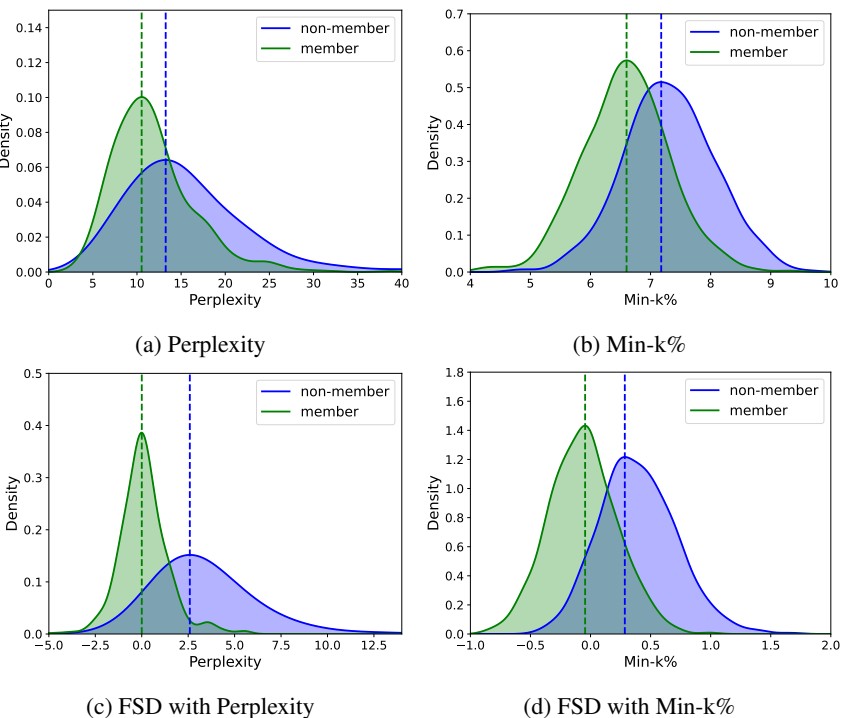

Figure 5: Distribution of scores from pre-trained model vs. FSD. We contrast the score distribution from the pre-trained model using perplexity and our FSD with perplexity(a & c). Similarly, we contrast the Min-k% scores distribution from the pre-trained model and our FSD (b & d). Using FSD leads to enlarging the gap between members and non-members.

**Can FSD improve the performance of detection methods based on scoring functions?** We compare the TPR@5%FPR score with our method and baselines on WikiMIA and ArXivTection datasets across various large language models in Table 12. The results show that our method significantly improves the TPR@5%FPR score of the baseline methods.

**Is FSD effective with different-sized models?** We verify the performance of baselines and our methods from different-sized LLaMA models (7B, 13B, 30B) on WikiMIA. In Table 13, we show the TPR@5%FPR score from different-sized LLaMA models. The results demonstrate that our method is effective with different-size models.

**Is our method affected by distribution difference?** We report the TPR@5%FPR score of baselines and our method on the original WikiMIA dataset, Deletion and Replacement. In Table 14, the results show that our method still improves the performance of baselines when mitigating the temporal shift between members and non-members.

### D.3 ADDITIONAL RESULTS

**The performance of our method on the Pile dataset** We also conduct experiments on the Pile dataset. Concretely, following prior work (Maini et al., 2024), we evaluate our methods on the twenty subsets of the Pile dataset. Here, the validation set of the pile dataset was not trained on the Pythia models (Biderman et al., 2023). Thus, we perform experiments on the Pythia-6.9B model, utilizing the training and validation sets as members and non-members, respectively. For each dataset, we randomly sample a few non-members with a sample ratio of 0.3 from the validation set for fine-tuning. Then, we evaluate our method on a balanced dataset composed of members and non-members. Notably, in our experiments, there is no overlap between the fine-tuning dataset and the evaluation data.

Table 13: TPR@5%FPR score for pretraining data detection with baselines and our method from the different-sized LLaMA model on WikiMIA. *Base* and *+Ours* respectively denote the baselines and our method. Bold shows the superior result.

| Method | LLaMA-7B | | LLaMA-13B | | LLaMA-30B | |
|---|---|---|---|---|---|---|
| | *Base* | *+Ours* | *Base* | *+Ours* | *Base* | *+Ours* |
| Perplexity | 0.09 | **0.41** | 0.11 | **0.61** | 0.15 | **0.40** |
| Zlib | 0.10 | **0.18** | 0.13 | **0.13** | 0.11 | **0.25** |
| Lowercase | 0.09 | **0.47** | 0.10 | **0.56** | 0.11 | **0.44** |
| MIN-K% | 0.15 | **0.25** | 0.18 | **0.26** | 0.19 | **0.20** |

Table 14: TPR@5%FPR score from the LLaMA-7B model with our method and baselines using the original WikiMIA, data removing timestamps (Deletion), and data replacing the year of timestamps with 2023 (Replacement). *Base* and *+Ours* denote the baseline methods and our method, respectively. **Bold** shows the superior result.

| Method | Origin | | Deletion | | Replacement | |
|---|---|---|---|---|---|---|
| | *Base* | *+Ours* | *Base* | *+Ours* | *Base* | *+Ours* |
| Perplexity | 0.09 | **0.41** | 0.13 | **0.23** | 0.04 | **0.12** |
| Lowercase | 0.10 | **0.18** | 0.06 | **0.13** | 0.03 | **0.15** |
| Zlib | 0.09 | **0.47** | 0.12 | **0.23** | **0.09** | 0.06 |
| MIN-K% | 0.15 | **0.25** | 0.10 | **0.14** | 0.04 | **0.07** |

In Table 15, the results show that our method improves the performance of baselines on most subsets of the Pile dataset under the Pythia-6.9B model. For example, our FSD improves the AUC score of the perplexity-based method from 0.528 to 0.885 on BookC2, a significant direct improvement of 67%. At the same time, our FSD improves the average AUC score of the perplexity-based method from 0.503 to 0.625 on the pile dataset, a notable direct improvement of 24.3%. This demonstrates the effectiveness of our method in the Pile dataset.

**Fine-tuning on BookC2 with varying data size**   We also conduct experiments on the BookC2 subset of the Pile dataset under the Pythia-6.9B model to investigate the effect of varying the fine-tuning data size on the pretraining data detection. Specifically, we randomly sample varying amounts of non-members (0, 50, 100, 150, 200, 250, 300, 350, 400, 450, 500) from the validation set of the BookC2 as fine-tuning datasets. In addition, we sample 1400 members and non-members from the train and validation sets of the BookC2 to construct a balanced test set of 2800 examples.

Figure 6a shows that our method achieves better performance as the size of the fine-tuning dataset increases. Notably, our method is highly data-efficient, achieving significant improvements with only a few non-members for fine-tuning. For instance, our method improves the AUC score of the Zlib method from 0.48 to 0.78, by leveraging only 100 non-member data for fine-tuning. In addition, the results of the TPR@5%FPR score show a similar trend, which can be found in Figure 6b.

**Fine-tuning using non-members from different domains**   Our method requires a few non-member data from a specific domain for fine-tuning. This raises a question: *how does our method perform when fine-tuned on non-member data from a different domain?* To investigate the performance of our method when fine-tuning using data from an unrelated domain. Firstly, we randomly sample 231 and 238 non-members from the WikiMIA and ArXivTection datasets to construct a fine-tuning dataset comprising a mix of domains. Then, we fine-tune the LLaMA-7B model on the constructed dataset and evaluate our method on WikiMIA and ArXivTection datasets.

Our results in Table 16 show that our method can also significantly improve the performance of baselines, indicating the effectiveness of our methods when fine-tuning with non-members from a mix of domains. We also evaluate our methods on ArXivTection while fine-tuning using non-members from WikiMIA. The results indicate that our method fails to improve the performance of

Table 15: AUC score for pretraining data detection with baselines and our method from the Pythia-6.9B on the Pile dataset. *Base* and ***+Ours*** respectively denote the baseline methods and our method. **Bold** shows the superior result.

| Method | Wiki | | BookC2 | | Gutenberg | | HackerNews | | Enron | |
| --- | --- | --- | --- | --- | --- | --- | --- | --- | --- | --- |
| | *Base* | *+Ours* | *Base* | *+Ours* | *Base* | *+Ours* | *Base* | *+Ours* | *Base* | *+Ours* |
| Perplexity | 0.471 | **0.614** | 0.528 | **0.885** | 0.528 | **0.661** | 0.471 | **0.565** | 0.510 | **0.678** |
| Lowercase | 0.466 | **0.626** | 0.518 | **0.725** | 0.546 | **0.551** | 0.450 | **0.512** | 0.484 | **0.659** |
| Zlib | 0.496 | **0.619** | 0.477 | **0.907** | 0.496 | **0.686** | 0.474 | **0.550** | 0.560 | **0.667** |
| MIN-K% | 0.512 | **0.611** | 0.510 | **0.841** | 0.536 | **0.612** | 0.498 | **0.535** | 0.570 | **0.646** |

| Method | CC | | arXiv | | Europarl | | FreeLaw | | GitHub | |
| --- | --- | --- | --- | --- | --- | --- | --- | --- | --- | --- |
| | *Base* | *+Ours* | *Base* | *+Ours* | *Base* | *+Ours* | *Base* | *+Ours* | *Base* | *+Ours* |
| Perplexity | 0.541 | **0.546** | **0.514** | 0.505 | 0.514 | **0.601** | 0.478 | **0.515** | 0.509 | **0.548** |
| Lowercase | 0.502 | **0.547** | 0.523 | **0.530** | 0.521 | **0.556** | 0.476 | **0.507** | 0.491 | **0.513** |
| Zlib | 0.529 | **0.576** | **0.540** | 0.505 | 0.462 | **0.609** | 0.492 | **0.503** | 0.491 | **0.562** |
| MIN-K% | **0.557** | 0.542 | **0.515** | 0.502 | 0.512 | **0.583** | 0.492 | **0.500** | 0.513 | **0.551** |

| Method | Books3 | | Nih | | OpenWebtext2 | | PhilPapers | | OpenSubtitles | |
| --- | --- | --- | --- | --- | --- | --- | --- | --- | --- | --- |
| | *Base* | *+Ours* | *Base* | *+Ours* | *Base* | *+Ours* | *Base* | *+Ours* | *Base* | *+Ours* |
| Perplexity | **0.560** | 0.509 | 0.463 | **0.599** | 0.490 | **0.580** | 0.571 | **0.869** | **0.525** | 0.521 |
| Lowercase | **0.550** | 0.524 | **0.608** | 0.512 | 0.486 | **0.547** | 0.633 | **0.718** | **0.538** | 0.528 |
| Zlib | 0.550 | **0.581** | 0.416 | **0.599** | 0.475 | **0.586** | 0.678 | **0.871** | **0.550** | 0.530 |
| MIN-K% | 0.552 | **0.554** | 0.463 | **0.560** | 0.510 | **0.567** | 0.606 | **0.826** | 0.525 | **0.535** |

| Method | StackExchange | | Math | | YoutubeSubtitles | | USPTO | | Ubuntu | |
| --- | --- | --- | --- | --- | --- | --- | --- | --- | --- | --- |
| | *Base* | *+Ours* | *Base* | *+Ours* | *Base* | *+Ours* | *Base* | *+Ours* | *Base* | *+Ours* |
| Perplexity | 0.640 | **0.678** | **0.530** | 0.504 | 0.392 | **0.756** | 0.537 | **0.606** | 0.282 | **0.767** |
| Lowercase | 0.579 | **0.641** | 0.508 | **0.513** | 0.495 | **0.546** | 0.510 | **0.582** | **0.496** | 0.476 |
| Zlib | 0.595 | **0.686** | **0.513** | 0.502 | 0.445 | **0.736** | 0.484 | **0.604** | 0.423 | **0.592** |
| MIN-K% | 0.637 | **0.670** | **0.524** | 0.510 | 0.380 | **0.692** | 0.549 | **0.596** | 0.329 | **0.561** |

Table 16: AUC score for pretraining data detection with baselines and our method on WikiMIA and ArXivTection under the LLaMA-7B. Wiki (Mix) denote evaluating on WikiMIA and fine-tuning using data from a mix of domains. ArXiv (Wiki) denote evaluating on ArXivTection and fine-tuning on WikiMIA. *Base* and ***+Ours*** respectively denote the baseline methods and our method.

| Method | Wiki (Mix) | | ArXiv (Mix) | | ArXiv (Wiki) | |
| --- | --- | --- | --- | --- | --- | --- |
| | *Base* | *+Ours* | *Base* | *+Ours* | *Base* | *+Ours* |
| Perplexity | 0.64 | **0.91** | 0.68 | **0.93** | **0.68** | 0.52 |
| Lowercase | 0.58 | **0.73** | 0.50 | **0.73** | 0.50 | 0.50 |
| Zlib | 0.62 | **0.91** | 0.57 | **0.92** | 0.57 | **0.64** |
| MIN-K% | 0.65 | **0.84** | 0.76 | **0.87** | 0.76 | 0.61 |

baselines, since the fine-tuning data comes from an entirely unrelated domain to the evaluation data.

**How do the fine-tuning parameters affect the performance of our method?**    To investigate the impact of different fine-tuning parameters on the performance of our method, we conduct experiments on the WikiMIA dataset with different fine-tuning parameters, including learning rate (e.g.

Table 17: AUC score of baselines and our method on WikiMIA under the LLaMA-7B with different fine-tuning parameters. *Base* and ***+Ours*** respectively denote the baseline methods and our method. **Bold** shows the superior result.

| **Method** | Learning Rate | | | | LoRA Rank | | | | Epoch | | | |
|---|---|---|---|---|---|---|---|---|---|---|---|---|
| | *Base* | $10^{-5}$ | $10^{-4}$ | $10^{-3}$ | *Base* | **8** | **16** | **32** | *Base* | **1** | **2** | **3** |
| Perplexity | 0.64 | 0.81 | 0.84 | **0.92** | 0.64 | 0.92 | 0.92 | 0.92 | 0.64 | 0.91 | 0.91 | **0.92** |
| Lowercase | 0.58 | 0.60 | 0.64 | **0.69** | 0.58 | 0.69 | 0.68 | 0.69 | 0.58 | 0.65 | 0.64 | **0.69** |
| Zlib | 0.62 | 0.73 | 0.78 | **0.90** | 0.62 | 0.91 | 0.90 | 0.90 | 0.62 | 0.87 | 0.87 | **0.90** |
| MIN-K% | 0.65 | 0.76 | 0.81 | **0.85** | 0.65 | 0.87 | 0.85 | 0.86 | 0.65 | 0.86 | **0.87** | 0.86 |

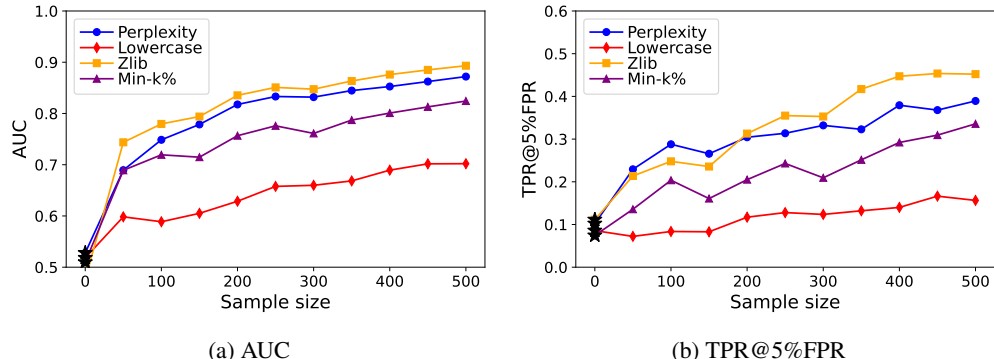

(a) AUC  (b) TPR@5%FPR

Figure 6: AUC and TPR@5%FPR of scoring functions with FSD, using auxiliary datasets with varying sizes. Notably, ★ represents the baseline without FSD.

1e-3, 1e-4, 1e-5), epoch (e.g. 1, 2, 3) and LoRA rank (e.g. 8, 16, 32). In Table 17, the results show that our method is relatively insensitive to LoRA rank and the number of fine-tuning epochs. However, considering the learning rate parameter, a learning rate of 0.001 enables our method to perform better.

