# OpenReview forum: "Fine-tuning can Help Detect Pretraining Data from Large Language Models"
_ICLR.cc/2025/Conference — ICLR 2025 Poster_

### Official Review · Reviewer_h5aY · 2024-10-18

**Soundness:** 3
**Presentation:** 4
**Contribution:** 3
**Rating:** 6
**Confidence:** 5

**Summary:**

This paper found that fine-tuning significantly reduces the perplexity of non-member data, which inspired the author to propose a new data detection method, FSD. The method determines whether data was used in training by comparing its perplexity in the original model versus the fine-tuned model. Experimental results showed a significant improvement over baseline methods.

**Strengths:**

1. The improvement is very significant.
2. The experiments and analysis are solid.
3. The writing is good and easy to follow.

**Weaknesses:**

1. This method requires fine-tuning the models. However, many models do not support fine-tuning especially white-box models, making this approach impractical in reality.
2. Fine-tuning every model you want to detect would be very costly.
3. Meta-evaluation typically fine-tunes a model on the data that needs to be detected, in order to assess whether the detection method is effective. However, this method is similar to the meta-evaluation.
4. The in-distribution non-member data is not easy to be acquired for fine-tuning.

**Questions:**

See weaknesses.

---

> ### Author Response · Authors · 2024-11-21
> **Response to Reviewer h5aY**
>
> Thanks for your recognition and the valuable suggestions. Please find our response below.
>
> ### **1. The setting is realistic in practical [W1, W4]**
> Thank you for raising the concerns. In this work, we assume the access to fine-tuning the pre-trained models and a few non-member data. We provide a detailed explanation of the two realistic assumptions below.
>
> * **Fine-tuning the models**:
> Our method assumes the access to fine-tune the pre-trained models. For the open-source models, we have the ability to fine-tune the models directly. In addition, many API-based commercial models provide the fine-tuning access via API interfaces, enabling task-specific customization, such as GPT-4o [1]. Therefore, our method can be broadly adopted for all open-source models and many popular commercial APIs, which is sufficient to support the development of such fine-tuning methods.
>
> * **Collecting non-member data**:
> Due to the vast availability of public text data, it is practical to collect non-member datasets in a special domain, by comparing the LLM release date with the data creation timestamp. The process for collecting data is described in Subsection 3.2. Our method can achieve significant results with only 100 unseen data, which makes it practical for real applications. When the non-member data is unavailable, an alternative solution is to fine-tune with member data in our framework, which can also improve the detection performance (but worse than using non-members).
>
> In summary, our method provides an effective solution to utilize non-member (or member) data to significantly improve the detection when the data is available. This highlights the contribution of our method in many real scenarios.
>
>
> ### **2. The computational cost of fine-tuning. [W2]**
> Thank you for raising the concerns. In fact, our method only requires **100** examples to achieve significant performance, and the finetuning can be completed in **109 seconds**. Thus, our method is high-efficiency with low computational overhead, making it practical for real applications. Please find the detailed analysis in the General Response.
>
> ### **3. Relations to meta-evaluation [W3]**
> Thank you for pointing out the potential confusion. We are not familiar with the area of meta-evaluation, so we provide the difference according to your description.
>
> 1. **Motivation**: our method employs fine-tuning to produce score deviations to differentiate members and non-members (members with low deviations). Meta-evaluation seems to use fine-tuning to learn the patterns of data that should be detected.
> 2. **Model usage**: we use fine-tuned models to calculate the score deviations as a new score for detecting pretraining data. Meta-evaluation uses fine-tuned models to evaluate the effectiveness of LLM evaluators.
> 3. **Data usage**(not sure): In this work, we fine-tune the model on a few collected non-members, which are not utilized to evaluate our method during the test phase. We believe this is different from meta-evaluation, which fine-tunes a model on all data that needs to be detected (non-members). In a word, we evaluate our method on examples that did not occur in the fine-tuning phase.
>
> We believe our method is significantly different from meta-evaluation, although both of them employ fine-tuning. Note that the explanation is based on your description, so the comparison might be not precise. If possible, could you share a related work of meta-evaluation, which fine-tunes a model on all data that needs to be detected?
>
> [1] https://platform.openai.com

---

> > ### Comment · Reviewer_h5aY · 2024-11-22
> > **Thank you for your reply**
> >
> > Your detailed reply addressed my concerns. I maintain my score and agree with accepting.

---

> > > ### Author Response · Authors · 2024-11-22
> > >
> > > Great thanks for your recognition.  We are glad that our response addressed your concerns, which also improves the quality of this work.

---

### Official Review · Reviewer_Zrvw · 2024-10-28

**Soundness:** 4
**Presentation:** 3
**Contribution:** 3
**Rating:** 8
**Confidence:** 3

**Summary:**

The paper tackles the issue of detecting pretraining data. It introduces a method where a pretrained model is fine-tuned with non-member data from the same domain. The goal is to increase the gap in scores like perplexity between member and non-member data, making it easier to spot training data.

**Strengths:**

* Clear introduction to the problem and motivation.
* The method is well explained and supported with intuitive examples, such as Figure 2.
* Comparisons with other relevant baselines, and showing significant improvement over them.
* Thorough experiments, including ablation studies that address additional research questions.
* Creative use of non-member data.

**Weaknesses:**

* The paper assumes access to unseen data in the same domain but doesn’t define 'domain' clearly. Could the authors explain how they handle differences between domains, especially if vocabularies differ a lot, and how they decide what data counts as 'same domain'?
* *"To the best of our knowledge, our method is the first to utilize some collected non-members in the task of pretraining data detection"* - There’s limited information on how to collect non-member data, which seems a key aspect of your method.
* The approach assumes access to both unseen data and model probabilities for each token, compared to other black-box assumption methods.

**Questions:**

* Could you explain more about the datasets, like how they’re labeled and how well they fit the models used? Is there a class imbalance in any of the datasets?
* Should fine-tuning happen for each example $x$ (or each group of examples $x_i$ within the same domain)? If so, that might be a big drawback.
* Fine-tuning can be delicate, and it’s easy to go past the optimal point. An experiment on how sensitive this method is to fine-tuning parameters (e.g., number of epochs, LoRA rank) would be useful.
* For a more diverse test set, what happens if we fine-tune once on a mix of domains?
* If non-member data in the specific domain is not available, is there any alternative in the general frame of this method?



**Minor suggestions:**
* If terms like "Data Contamination" or "Membership Inference Attack" are central, it might help to introduce them earlier (e.g., in the Introduction or Background). If not, why are they in the Related Work section?
* Adding an ethics statement could underscore the method's role in detecting sensitive information used in LLM training, addressing the ethical significance of such methodologies.
* Consider renaming either the “Method” section or subsection for clarity.
* Start a distinct “Results” section.

---

> ### Author Response · Authors · 2024-11-21
> **Response to Reviewer Zrvw  (1/2)**
>
> Thanks for your recognition and the valuable suggestions. Please find our response below.
>
> ### **1. Definition of 'domain' [W1]**
> Thank you for pointing out the potential confusion due to insufficient descriptions.
> In the literature on NLP, the term "domain" typically refers to the distinct areas or fields where language is used [1], such as different topics (i.e., business news, medical knowledge, and legal documents). This is universally used to denote some coherent data related to a topic, style or genre [2]. Researchers usually predetermine the domain of the given dataset in some NLP tasks, such as domain adaptation[3] and domain-specific fine-tuning [9]. In this work, we classify domains based on topics (e.g., event data from Wikipedia and arXiv research papers). In practice, we can use a topic from the same source to denote the specific domain.
>
> Here, we provide a simple notation for reference. Let $D = $ {$D_1, D_2, \dots, D_n$ } denotes the large-scale pretraining dataset, which is comprised of data from multiple distinct domains. Where $D = $ {$X_i$} denotes a specific domain within the pretraining dataset. $X_{i}$ represents the text corpus that are sampled from the underlying distribution of the domain $P_{X_{i}}$.
>
>
> ### **2. How to collect non-member data [W2]**
> Thank you for the suggestion. In practice, developers can collect non-member data by comparing the LLM release date and data creation timestamp (as described in Subsection 3.2), following previous works [4]. For example, given a task that determines whether some texts from Wikipedia are used for pretraining by LLaMA, we can collect a few events that occurred after February 2023 from Wikipedia as a non-member dataset for fine-tuning. This is because data created after February 2023 are unseen by LLaMA, as the LLaMA was released in February 2023. Note that our method can achieve significant results with only 100 unseen data, so it is realistic to apply our method in real applications. In addition, our framework can also applied to member data if a few of them are available, as shown in the Discussion.
>
> ### **3. The realistic assumptions of our method [W3]**
> Yes, our method requires unseen data and model probabilities for each token. We clarify that this method is designed to improve the current likelihood-based methods [4,5,6], which assume access to the output probabilities of LLMs. Despite the assumption, those methods are useful and valuable in practice, because **the access is realistic for open-sourced LLMs and many commercial APIs**, such as GPT-4o. Notably, our method provides an effective solution to exploit non-member data for significant improvement, when a few data is available. This is a unique contribution to this work, and collecting a few non-member data is also realistic in many scenarios.
>
> ### **4. Details about datasets [Q1]**
>
> Thank you for the question. In our work, we conduct experiments on four datasets, including WikiMIA [4], ArXivTection [7], BookTection [7], and BookMIA [4]. For the label of membership, the dataset creators use data occurring before LLM's release date as members, and vice versa [8]. This setting is generally adopted in pretraining data detection of LLMs[4,6,7,10]. In addition, the members and non-members in these datasets are basically balanced, as shown in Appendix C.1. The details of the datasets are presented in Appendix B.
>
> In addition, we conduct experiments on datasets that reduce the temporal difference in Discussion. we implement two strategies to mitigate the temporal shift in the dataset: (1) removing timestamps in the dataset (Deletion) and (2) replacing the year of
> timestamps with 2023 in the dataset (Replacement). As suggested by reviewers, we also add new results on datasets with IID setups [11], which is introduced in the General Response.
>
> ### **5. Fine-tuning does not happen for each example [Q2]**
> Thank you for the great question. In our experiments, there is **no overlap** between the fine-tuning dataset and the evaluation data. Therefore, we do not perform fine-tuning on each sample. Specifically, we only collect a few non-members to build the dataset for fine-tuning and these data are not used for evaluation. In addition, our method can achieve significant results with only 100 collected data, as shown in Figure 4.

---

> ### Author Response · Authors · 2024-11-21
> **Response to Reviewer Zrvw (2/2)**
>
> ### **6. Analysis of different fine-tuning parameters [Q3]**
> Thank you for the suggestion. As suggested, we conduct experiments on the WikiMIA dataset with different fine-tuning parameters, including learning rate (e.g. 1e-3, 1e-4, 1e-5), epoch (e.g. 1, 2, 3) and LoRA rank (e.g. 8, 16, 32). The table below presents the AUC score for pretraining data detection with baselines and our method under the LLaMA-7B model. The results show that our method is relatively insensitive to LoRA rank and the number of fine-tuning epochs, while a learning rate of 0.001 performs much better than other choices. We add the sensitivity analysis of hyperparameters in Appendix D.3 of the revised version.
>
> | Type       |      |      |      | learning rate |      |      |      | LoRA Rank |      |      |          | Epoch    |
> | ---------- | ---- | ---- | ---- | ------------- | ---- | ---- | ---- | --------- | ---- | ---- | -------- | -------- |
> | Method     | Base | 10−5 | 10-4 | 10-3          | Base | 8    | 16   | 32        | Base | 1    | 2        | 3        |
> | Perplexity | 0.64 | 0.81 | 0.84 | **0.92**      | 0.64 | 0.92 | 0.92 | 0.92      | 0.64 | 0.91 | 0.91     | **0.92** |
> | Lowercase  | 0.58 | 0.60 | 0.64 | **0.69**      | 0.58 | 0.69 | 0.68 | 0.69      | 0.58 | 0.65 | 0.64     | **0.69** |
> | Zlib       | 0.62 | 0.73 | 0.78 | **0.90**      | 0.62 | 0.91 | 0.90 | 0.90      | 0.62 | 0.87 | 0.87     | **0.90** |
> | MIN-K%     | 0.65 | 0.76 | 0.81 | **0.85**      | 0.65 | 0.87 | 0.85 | 0.86      | 0.65 | 0.86 | **0.87** | 0.86     |
>
> ### **7. Fine-tuning on a mix of domains [Q4]**
> Thank you for the suggestion. Please find the detailed analysis in the General Response.
>
> ### **8. An alternative in the fine-tuning framework [Q5]**
> Thanks for the question. In the Discussion, we show that fine-tuning with **members** can also provide some improvements to some extent. While it cannot achieve comparable performance to fine-tuning with non-members, utilizing a few members can be an effective alternative when the non-members are unavailable.
>
> ### **9. Minor suggestions**
> Thank you for the suggestions. We have fixed these in the updated version.
>
> [1] Ramponi A, et al. Unsupervised domain adaptation in nlp—a survey. ICCL2020.
>
> [2] Plank B. What to do about non-standard (or non-canonical) language in NLP. KONVENS2016.
>
> [3] Wu H, et al. Domain-Adaptive pretraining methods for dialogue understanding. ACL2021.
>
> [4] Weijia S, et al. Detecting pretraining data from large language models. ICLR2024.
>
> [5] Yonatan O, et al. Proving test set contamination in black-box language models. ICLR2024.
>
> [6] Wentao Y, et al. Data contamination calibration for black-box llms. ACL2024.
>
> [7] Duarte A V, et al. De-cop: Detecting copyrighted content in language models training data. ICML2024.
>
> [8] Gao L, et al. The pile: an 800gb dataset of diverse text for language modeling. arXiv:2101.00027, 2020.
>
> [9] Zheng J, et al. Fine-tuning large language models for domain-specific machine translation. arXiv:2402.15061, 2024.
>
> [10] Zhang W, et al. Pretraining data detection for large language models: a divergence-based calibration method. EMNLP2024.
>
> [11] Maini P, et al. LLM Dataset Inference: did you train on my dataset? arXiv:2406.06443, 2024.

---

### Official Review · Reviewer_J4zF · 2024-11-03

**Soundness:** 3
**Presentation:** 2
**Contribution:** 2
**Rating:** 6
**Confidence:** 3

**Summary:**

The paper proposes a new method for improving the accuracy of pretraining data detection in LLMs, called Fine-tuned Score Deviation. they shows that fine-tuning models on a small set of non-member data increases the deviation between score distributions of seen and unseen data. They validate this claim via experiments on WikiMIA, BookMIA, ArXivTection and BookTection datasets. FSD consistently improves the performance of existing detection methods based on scoring functions like perplexity and Min-k%.

**Strengths:**

Very interesting observation, and notable improvement in pretraining data detection accuracy. I think this paper has a clearly defined objective, and interesting empirical results.

The results are strong, showing substantial improvements in AUC and tpr at low fpr across the selected datasets and models. FSD helps improving existing score-based pretraining data detection methods.

**Weaknesses:**

1. Have you experimented on the MIMIR dataset? This dataset seems to be more challenging, and authors claim this is because seen and unseen examples are not from different temporal distributions. I am very interested to see how this method helps MIA on MIMIR dataset.

2. They have not explained or given intuition about why finetuning increases the gap between distributions.

3.  The method requires fine-tuning the LLM, which makes it more expensive compared to existing methods. If authors find some intuition about why finetuning has this impact, they may find less expensive methods.

**Questions:**

1. Do you have intuitions or theoretical understanding about why finetuning increases the gap between distributions?

2. What is the role of the finetuning technique you use? Is there a dependency on LORA?

3. How sensitive is FSD to using unseen data from a loosely related domain impact detection performance?

---

> ### Author Response · Authors · 2024-11-21
> **Response to Reviewer J4zF**
>
> Thanks for your recognition and the valuable suggestions. Please find our response below.
>
> ### **1. Results on IID setups [W1]**
> Thank you for the suggestion. Reviewer oZEC also mentioned concerns about the temporal shifts and suggest to conduct experiments on data from the Pile with IID setups. As reviewers suggested, we present new results on three datasets that are built from the Pile. The results are shown in the General Response.
>
> ### **2. Clarification of intuition [W2, Q1]**
> There might be some misunderstandings. We do **not** claim that "fine-tuning increases the gap between distributions". We clarify that our method utilizes fine-tuning to compute the FSD score, which reflects a large gap between members and non-members. In particular, the FSD score measures the deviation of **perplexity (or other existing scores)** between the pre-trained model and the fine-tuned model. This is motivated by the phenomenon shown in Figure 2: finetuning on a few unseen data decreases the perplexity of non-members but almost keeps those of members unchanged. The intuition is that members already have a low perplexity, which cannot be easily decreased via finetuning. Thus, we utilize the difference in score deviation to provide an effective metric for detection.
>
> Notably, **finetuning does not increase the gap between members and non-members** (even closer, as shown in Figure 2). Instead, our method measures the score deviation of the same examples after fine-tuning.
>
> ### **3. The computational cost of our method [W3]**
> Thank you for the great suggestion. We agree that finding a less expensive method is an interesting direction in the future. However, it does not affect the contribution of this work: revealing the phenomenon and proposing an effective method accordingly. In fact, our method only requires **100** examples to achieve significant performance, and the finetuning can be completed in **109 seconds**. Thus, our method is high-efficiency with low computational overhead, making it practical for real applications. Please find the detailed analysis in the General Response.
>
> ### **4. Our method supports various fine-tuning techniques [Q2]**
> Thank you for the question. Indeed, we investigate the effect of different fine-tuning methods in our paper, which is discussed in **Section 5**. We conduct experiments with **3** fine-tuning methods to fine-tune LLaMA-7B, respectively. The results show that our method can be implemented with different fine-tuning methods and does not rely on LoRA.
>
> ### **5. Finetuning using non-members from a loosely related domain [Q3]**
> Thank you for the suggestion. Please find the detailed analysis in the General Response.

---

> > ### Comment · Reviewer_J4zF · 2024-11-24
> >
> > Thanks for the clarifications and additional results. Why not testing on the MIMIR dataset?

---

> > > ### Author Response · Authors · 2024-11-25
> > >
> > > Thank you for the reply. Due to the time limit, we provided the results on three subsets of Pile in the general response. As suggested by your reply, we also conduct experiments on the MIMIR dataset today. We find that current perplexity-based methods do not work on MIMIR and our method also fails to improve their performance effectively.
> > >
> > > To explain this failure, we carefully check the dataset and the score deviations during the finetuning. We find that finetuning on non-members sampled from MIMIR cannot decrease (and may even increase) the perplexities of both members and non-members. Generally, finetuning on subsets should decrease the perplexities of data from the same distribution. We conjecture that **our method fails to conduct effective finetuning due to the poor quality of this dataset**, which is largely different from the training distribution. The texts in MIMIR generally have ambiguous meanings, resulting in difficulty in computing meaningful perplexities (see examples from [the anonymous link](https://anonymous.4open.science/r/Sample-A7C6/example.md)). Thus, finetuning is not applicable to such a dataset with poor data quality, which may also explain the failure of the perplexity-based method. We add this as a drawback of fine-tuning-based methods in the limitation of the revised version.

---

> > > > ### Comment · Reviewer_J4zF · 2024-11-26
> > > >
> > > > Thanks for checking the dataset and addressing this limitation. It addresses my points. I think the paper has extensive experiments now.

---

> > > > > ### Author Response · Authors · 2024-11-26
> > > > >
> > > > > Thank you so much for the recognition. We are glad that our response addressed your concerns. Your feedback is highly valuable in improving the quality of this work.

---

### Official Review · Reviewer_oZEC · 2024-11-04

**Soundness:** 2
**Presentation:** 4
**Contribution:** 2
**Rating:** 5
**Confidence:** 5

**Summary:**

The paper addresses the problem of detecting prestrainign data from LLMs. They notice that when fine-tuning an LLM on a few non-members selected by a cut-off date, the perplexity on other non-members (also based on the same cut-off date) decrease. Based on this observation, the authors propose a new method (Fine-tuned Score Deviation; FSD) to improve prior membership inference attacks (MIA) by adjusting the scores based on this fine-tuning phenomenon. They conduct experiments on 4 cut-off based benchmarks: WikiMIA, BookMIA, ArXivTection and BookTection and show significant gains wrt the baselines. The specific questions they tackle are:

1. Can FSD improve the performance of current MIA methods?
2. How many samples does the method need for fine-tuning?
3. Does FSD work across different model sizes?
4. Does FSD work with members for fine-tuning, instead of non-members?
5. Does FSD work for any fine-tuning method?

**Strengths:**

1. The idea is simple and intriguing, enlarging the difference between members and non-members could boost any MIA method.
2. The idea works for any MIA method.

**Weaknesses:**

The evaluation has major flaw. It only uses benchmarks that separates members and non-members based on cut-off dates. As Duan et al., 2024, Das et al., 2024, and Maini et al., 2024 show, this is fundamentally wrong because it introduces a temporal shift that bias the benchmark. These works have consistently proved the need to avoid evaluating MIA based on cut-off dates. This paper acknowledges these works on lines 458-459 and questions whether the results of this ICLR submission are influenced by this temporal shift. However, instead of following the recommendations from these papers for a correct evaluation, they simply remove timestamps. This is not enough as Duan et al., (2024) shows. Duan et al., shows that the cut-offs introduces changes in the n-grams. An IID split have a higher n-gram overlap between members and non-members than the temporal split. Therefore, even when removing the timestamps in the plain texts, the n-gram distribution is different, making the evaluated method classify temporal data rather than membership. Moreover, the authors acknowledge this possibility in lines 535-537 but they do not follow the recommendations of the papers mentioned above, which are to evaluate on for example Pythia with data from the Pile.

**Questions:**

* Please conduct your experiments on an IID setup (see the works of Duan et al., 2024, Das et al., 2024, and Maini et al., 2024 that you cite; eg: using Pythia use as members the training set and as non-members the dev and test sets) so that we can clearly see the effectiveness of your work.
* It is possible that the results of Figure 4 are explained by the temporal bias of the evaluation. Training more on new data might make the method identify better the n-gram distribution of the documents after the cut-off. We can reject this hypothesis by evaluating on an IID setup.

---

> ### Author Response · Authors · 2024-11-21
> **Response to Reviewer oZEC**
>
> Thanks for your recognition and the valuable suggestions. Please find our response below.
>
> ### **1. Results of IID setup [W1, Q1]**
> Thank you for the suggesion. As suggested, we conduct experiments on data from the Pile dataset, with an IID setup [1]. We present the results in the General Response and add it in Appendix D.3.
>
> ### **2. Ablation study with an IID setup [Q2]**
> Thank you for the great question. To evaluate our method on an IID setup, we conduct experiments on the BookC2 subset of the Pile dataset under the Pythia-6.9B model. Specifically, we randomly sample varying amounts of non-members (0, 50, 100, 150, 200, 250, 300, 350, 400, 450, 500) from the validation set of the BookC2 as fine-tuning datasets. In addition, we sample 1400 members and 1400 non-members from the train and validation sets of the BookC2 to construct a balanced test set of 2800 examples. Importantly, the fine-tuning data are strictly excluded from the evaluation in the testing phase, ensuring unbiased results.
>
> In Appendix D.3, we present the results of fine-tuning on auxiliary datasets with varying sizes. The results show that **our method achieves better performance as the size of the fine-tuning dataset increases**. Notably, our method is highly data-efficient, achieving significant improvements with only a few non-members for fine-tuning. For instance, our method improves the AUC score of the Zlib method from 0.48 to 0.78, by leveraging only **100** non-member data for fine-tuning.
>
> [1] Maini P, et al. LLM Dataset Inference: did you train on my dataset? arXiv:2406.06443, 2024.

---

> > ### Comment · Reviewer_oZEC · 2024-11-26
> >
> > Thank you for conducting experiments on The Pile. The results seem promising. I have some questions about your experimental setup. You claim to use varying amounts of non-members (up to 500) from the validation set of BookC2. However, there are only 26 documents on the validation set. Did you chunk them to have more non-members? What is the context window size you use?
> >
> > Even though these results are promising, most experiments are unfortunately conducted using a flawed benchmark, as Duan et al., 2024, Das et al., 2024, and Maini et al., 2024 have shown. Therefore, we do not have conclusive evidence of the effectiveness of this method until full experiments on Pythia models (or GPT-Neo) on The Pile are conducted.

---

> > > ### Author Response · Authors · 2024-11-27
> > >
> > > Thank you for the reply. Please find our response below.
> > >
> > > ### **1. Details of datasets**
> > > Yes, the text is chunked in BookC2. Following your suggestion, we use the datasets provided by the previous work [1]. They preprocess these datasets by chunking the texts to more examples, as described in Section 5.3 of their paper. We also check their code and find that the context window size is 512 by default. Notably, there is no overlap between the samples (as shown in [their code](https://github.com/pratyushmaini/llm_dataset_inference/blob/main/dataloader.py)).
> > >
> > > ### **2. Full experiments on the Pile dataset**
> > > Thank you for the suggestion. As suggested, we conduct full experiments on the 20 datasets from Pile [1] with the Pythia-6.9B model. We present the detailed results of all datasets in Appendix D.3. The table below illustrates the **average** AUC scores of baselines and our method over the 20 subsets of Pile. The results show that **our method significantly improves the average detection performance**, while previous methods cannot provide meaningful performance. For instance, our method improves the average AUC score of the perplexity-based method from 0.503 to 0.625, a notable direct improvement of **24.3%**. Besides, our method can make significant improvements on **16**/20 subsets from Pile. This demonstrates the effectiveness of our method in the IID setup.
> > >
> > >
> > > | Method    | Perplexity | Lowercase | Zlib  | MIN-K% |
> > > | --------- | ---------- | --------- | ----- | ------ |
> > > | Base      | 0.503      | 0.519     | 0.507 | 0.515  |
> > > | **+Ours** | **0.625**      | **0.566**     | **0.624** | **0.600**  |
> > >
> > > [1] Maini P, et al. LLM Dataset Inference: did you train on my dataset? arXiv:2406.06443, 2024.

---

> > > ### Author Response · Authors · 2024-11-29
> > >
> > > Hi Reviewer oZEC,
> > >
> > > We apologize for any inconvenience this may cause. We are writing to follow up on our previous response, in case you missed it. As per your suggestion, we have conducted full experiments on **20 datasets** from Pile. The results demonstrate that our method performs well on most of them, providing conclusive evidence of its effectiveness. Besides, **All** other reviewers have agreed that our experiments are extensive, solid, and thorough. We look forward to your feedback and are available to discuss any remaining concerns you may have.
> > >
> > > Best regards,
> > >
> > > Authors of submission 1584

---

> > > > ### Comment · Reviewer_oZEC · 2024-12-02
> > > >
> > > > Dear authors,
> > > >
> > > > Thank you for your conducting experiments on all the datasets from the Pile. Now we have real evidence of your claim. Therefore, I am increasing the scores. However, the only experiments on a fair benchmark are conducted in the appendix. All the experiments in the main paper are conducted with flawed benchmarks for MIA (Duan et al., 2024, Das et al., 2024, and Maini et al., 2024). Therefore, I cannot increase the scores more.

---

> ### Author Response · Authors · 2024-12-02
>
> Hi Reviewer oZEC,
>
> Thank you for the reply and raising the score. Now all reviewers recognize the significance and contribution of this work. In the experiments of the current version, we use 4 datasets with risk of temporal shift and 20 datasets with the IID setup, which is extensive and solid.
>
> For the concern of presentation, we will **definitely** place the average results of 20 datasets from Pile in the **main paper of the final version**, and **emphasize** the importance of these datasets. This is not updated in the discussion, as we were waiting for the feedback of the reviewer, in case of any other suggestions on experiments. We will accomplish this in the final version, because the deadline of uploading a revised PDF was due before the reviewer's response.
>
> Thank you again for your efforts on improving this work.
>
> Best wishes,
>
> Authors

---

### Author Response · Authors · 2024-11-21
**General Response  (2/2)**

### **2. Minimal computational overhead with significant improvements.**
We find that several reviewers concern about the cost of fine-tuning. Here, we provide a detailed analysis to show the computational cost of our method.

* **Significant improvements with a few samples for fine-tuning.** In our work, we investigate how the size of fine-tuning data affects the performance of FSD, in Subsection 4.2. The results show that our method is highly data-efficient. For instance,  our method improves the AUC score of the perplexity-based method **from 0.63 to 0.91** on WikiMIA, by leveraging only **100 non-member data** for fine-tuning – a significant direct improvement of 44%. Thus,  Our method only needs a small number of samples for fine-tuning to improve detection, which requires low computational cost.

* **Fine-tuning with efficient parameter fine-tuning method.** Our method employs LoRA [1] to fine-tune the pre-trained model, achieving this with minimal computational cost. For example, we conducted a fine-tuning process on the LLaMA-7B model utilizing 230 samples from the WikiMIA dataset, supported by two NVIDIA L40 GPUs. With a batch size of 8 and 3 epochs, the fine-tuning is finished in only **109 seconds**, demonstrating its minimal computational overhead. Besides, our method does not rely on the LoRA and can work well with other PEFT, as shown in Section 5.

In summary, the computational cost of our method is acceptable in practice and can achieve dramatic improvements with minimal computational overhead.

[1] Hu, et al. Lora: Low-rank adaptation of large language models. ICLR 2022.


### **3. Fine-tuning using non-members from different domains**
We note that two reviewers are interested in our method's performance when fine-tuning data from a mix of domains. Thanks for the great suggestion. We add two experiments to explore the performance of our methods when fine-tuning using non-members from different domains.

* **Fine-tuning using non-members from a mix of domains**
Specifically, we randomly sample 231 and 238 non-members from the WikiMIA and ArXivTection datasets to construct a fine-tuning dataset comprising a mix of domains. The remaining part of the dataset is used for testing. Then, we fine-tune the LLaMA-7B model on the constructed dataset and evaluate our method on the two datasets, respectively. The results show that our method can also **significantly improve** the performance of baselines, indicating the effectiveness of our methods when fine-tuning with non-member data from a mix of domains.

| Dataset    | WikiMIA |           | ArXivTection |           |
| ---------- | ------- | --------- | ------------ | --------- |
| Method     | Base    | **+Ours** | Base         | **+Ours** |
| Perplexity | 0.64    | **0.91**  | 0.68         | **0.93**  |
| Lowercase  | 0.58    | **0.73**  | 0.50         | **0.73**  |
| Zlib       | 0.62    | **0.91**  | 0.57         | **0.92**  |
| MIN-K%     | 0.65    | **0.84**  | 0.76         | **0.87**  |


* **Fine-tuning using non-members from entirely unrelated domains**
Specifically, we fine-tune the LLaMA-7B model on non-members sampled from the ArXivTection dataset. Subsequently, we evaluate the performance of our method on the WikiMIA dataset for pretraining data detection. The table below shows the AUC scores of baselines and our method. The results indicate that our method **fails to improve** the performance of baselines, since the fine-tuning data comes from an entirely unrelated domain to the evaluation data.


| Method    | Perplexity | Lowercase | Zlib | MIN-K% |
| --------- | ---------- | --------- | ---- | ------ |
| Base      | **0.68**   | **0.50**  |  0.57|  **0.76**|
| **+Ours** | 0.52       | 0.50      |  **0.64**| 0.61 |


In summary, our method can work on a mix of domains, but fails to improve given data from unrelated domains. Therefore, our method requires to collect a few non-members that belong to the related domain, as highlighted in our work.

---

### Author Response · Authors · 2024-11-21
**General Response (1/2)**

We appreciate the reviewers' thoughtful feedback and valuable comments on our work. We are encouraged that reviewers recognize that the idea of our work is **intriguing** (oZEC), and point out that our work is **very interesting** (J4zF) and **creative** by leveraging non-member data (Zrvw). The reviewer finds that our method is **well explained** and **supported** with intuitive examples (Zrvw). We are pleased that reviewers find the experiments and analysis are **solid** (h5aY), and the empirical results are **interesting** and **strong** (J4zF) with **notable/significant** improvements (J4zF, Zrvw, h5aY). Besides, reviewers recognize that the writing is **good** and **easy to follow** (h5aY), with a **clearly defined objective** (J4zF) and a **clear introduction** to the problem and motivation (Zrvw).

The reviews allow us to strengthen our manuscript, and the changes are summarized below:

* Added introduction of “Membership Inference Attack” and “Data Contamination” in **Line 108-110**. [Zrvw]
* Revised the title of Section 3 in **Line 162**. [Zrvw]
* Revised the title of Subsection 3.2  **in Line 205**. [Zrvw]
* Added experiments on the pile dataset in **Line 346-347** and **Appendix D.3**. [oZEC, J4zF]
* Added experiments on fine-tuning with varying data size in **Line 360-362** and **Appendix D.3**. [oZEC]
* Added experiments on fine-tuning with different parameters in **Line 498-499** and **Appendix D.3**. [Zrvw]
* Added ethical statement in **Line 521-526**. [Zrvw]
* Added experiments on fine-tuning using data from different domains in **Appendix D.3**. [J4zF, Zrvw]
* Revised description on fine-tuned score deviation **in Line 077**.
---
¹ For clarity, we highlight the revised part of the manuscript in **blue** color.

In the following responses, we have addressed the reviewers' comments and concerns point by point and are willing to discuss any concerns you may have. Here, we provide responses to some common concerns of the reviewers.

### **1. Experiments on the Pile dataset**
We note that two reviewers are interested in the effectiveness of our method on a more challenging dataset. Here, we provide detailed results on three datasets from the Pile dataset that satisfies the IID setup [1].

Concretely, we evaluate our methods on the three subsets of the Pile dataset, following the prior work [2]. Here, the validation set of the pile dataset was not trained on the Pythia models [3]. Thus, we conduct experiments on the Pythia-6.9B model, utilizing the training and validation sets as members and non-members, respectively. For each dataset, we randomly sample a few non-members with a sample ratio of 0.3 from the validation set for fine-tuning. Then, we evaluate our method on a balanced test set composed of 1400 members and 1400 non-members.

The table below shows that our method **significantly improves the performance** of baselines on the Pile dataset under the Pythia-6.9B model. For example, our FSD improves the AUC score of the perplexity-based method from 0.528 to 0.885 on BookC2, a significant direct improvement of 67\%. This demonstrates the effectiveness of our method in the IID setup.

| Dataset    | Wiki  |           | BookC2 |           | Gutenberg |           |
| ---------- | ----- | --------- | ------ | --------- | --------- | --------- |
| Method     | Base  | **+Ours** | Base   | **+Ours** | Base      | **+Ours** |
| Perplexity | 0.471 | **0.614** | 0.528  | **0.885** | 0.528     | **0.661** |
| Lowercase  | 0.466 | **0.626** | 0.518  | **0.725** | 0.546     | **0.551** |
| Zlib       | 0.496 | **0.619** | 0.477  | **0.907** | 0.496     | **0.686** |
| MIN-K%     | 0.512 | **0.611** | 0.510  | **0.841** | 0.536     | **0.612** |

[1] Gao L, et al. The pile: an 800gb dataset of diverse text for language modeling. arXiv:2101.00027, 2020.

[2] Maini P, et al. LLM Dataset Inference: did you train on my dataset? arXiv:2406.06443, 2024.

[3] Stella B, et al. Pythia: a suite for analyzing large language models across training and scaling. ICML2023.

---

### Meta-Review · Area_Chair_9Vbo · 2024-12-16

**Metareview:**

This paper introduces a new method called Fine-tuned Score Deviation (FSD) to improve the performance of various scoring functions for pretraining data detection. The proposed method is inspired by an observation: after fine-tuning with a small amount of previously unseen data, the perplexities of LLMs perform differently for members and non-members. This observation is interesting and the method is technically sound. However, one highly confident reviewer raised a major concern about the datasets based on cut-off dates used in the experiments, and previous studies have shown that such datasets can be heavily biased, so they cannot well validate the performance of the proposed method. The authors attempted to address this issue by reporting new experimental results on additional datasets in the author's rebuttal. Another important issue is that the method requires fine-tuning the LLM, which makes it much more expensive compared to existing methods. I have mixed feelings about this paper and it is a borderline one.

**Additional Comments On Reviewer Discussion:**

The reviewers participated in the discussion and most of them increased the scores.  Reviewer oZEC raised a major concern about the datasets used in the evaluation, and the authors reported new experimental results on additional datasets. Reviewer oZEC still has a slightly negative attitude towards this paper.

---

### Decision · Program_Chairs · 2025-01-22

Accept (Poster)